# Biostimulants Promote Plant Development, Crop Productivity, and Fruit Quality of Protected Strawberries

**Veedaa Soltaniband, Annie Brégard, Linda Gaudreau and Martine Dorais ***

Centre de Recherche et d'Innovation sur les Végétaux, Département de Phytologie, Université Laval, Québec, QC G1V 0A6, Canada; veedaa.soltaniband.1@ulaval.ca (V.S.); annie.bregard@fsaa.ulaval.ca (A.B.); linda.gaudreau@fsaa.ulaval.ca (L.G.)
* Correspondence: martine.dorais@fsaa.ulaval.ca; Tel.: +1-418-6562131 (ext. 403939)

**Abstract:** Berries such as strawberries are recognized as a significant constituent of healthy human diets owing to their bioactive secondary metabolites. To improve crop sustainability, yield and berry quality, alternative production systems should be proposed such as organic farming and the use of biostimulants. Thus, we have compared within a complete randomized block design seven biostimulant treatments: 1-control, 2-seaweed extract, 3-*Trichoderma*, 4-mycorrhiza, 5-mixture of three bacteria, 6-combination of mycorrhiza + bacteria, and 7-citric acid. Strawberry plants were grown in conventional high tunnel (CH), conventional greenhouse (CG) and organic greenhouse (OG). Our results showed that biostimulants did not impact the soil microbial activity (FDA) when compared with the control. Leaf chlorophyll content and photosynthetic leaf performance were not affected by any studied biostimulants. Bacteria, citric acid, and the combination of mycorrhiza + bacteria increased the number of flowering stalks compared with the control in CH, while bacteria increased the crown diameter and all biostimulants increased fresh and dry shoot plant biomass. Citric acid increased leaf Ca content in CG, when all biostimulants increased leaf N content in CH. Studied biostimulants increased berry productivity in CH, while citric acid treatment had the highest yield in CG. The anthocyanins content increased with the use of biostimulants in CH, whereas *Trichoderma* (CG) and the combination of mycorrhiza + bacteria (OG) increased the Brix, total polyphenols, and anthocyanin contents of the berries compared with the control.

**Keywords:** strawberry; seaweed; *Trichoderma*; mycorrhiza; bacteria; citric acid; yield; Brix; polyphenols; anthocyanins

## 1. Introduction

Fruit production accounted for 20% of overall edible horticulture cash receipts in Canada, with a farm gate value of 1.2 billion dollars, which increased by 1% from 2020 to 2021 [1]. Among fruits, berries are recognized as a significant constituent of healthy human diets owing to their bioactive secondary metabolites and their potential use in the prevention of chronic diseases, such as cardiovascular diseases and cancer [2–4]. For growers, the critical aspect of strawberry production is achieving a fair marketable yield and profitability. Thus, conventional methods, such as using chemical fertilizers, pesticides, and soil fumigants, are largely used. However, strawberry was reported as one of the fruits with the highest level of pesticide residues, which might have a significant impact on the health of the population, particularly young children who love strawberries. It has also been reported that conventional practices have several adverse effects on the ecosystem [5]. Besides the negative effects of pesticides on human health and the environment, they are expensive and inefficient in several cases due to pesticide resistance or misuse.

On the other hand, from 2017 to 2020, the demand for organic food in Canada expanded by 23.4%, a fourfold increase, with 39% represented by fruits and vegetables [6].

Moreover, it was reported that the premium of organic strawberries can be, on average, 64% higher than the conventional price, with the premium reaching 308% during the fall or winter seasons [7], constituting interesting market opportunities for growers. However, the organic farming of strawberries faces several issues, such as pests and disease control, as well as soil nutrient balance and nutrient availability, that often limit crop productivity [8]. Therefore, in order to achieve a sustainable yield of healthy fruits, alternative approaches for conventional growers are needed to reduce chemical inputs without reducing crop productivity, while, for organic growers, new tools need to be proposed to improve the nutrient use efficiency and plant resilience to abiotic or biotic stresses [8,9].

An agroecological tool to improve crop productivity and plant resilience to stresses related to climate change would be the use of biostimulants, as proposed by several authors [9–13]. The European Biostimulant Industry Council (EBIC) [14] described biostimulants as "*organic or natural material products obtained from bioactive materials and/or microorganisms that can boost several molecular and physiological processes*". Therefore, biostimulants are applied exogenously and are considered as substances for plants to boost crop productivity, and improve the nutrient use efficiency and quality attributes, regardless of their nutrient content [13,15]. They can improve plant resilience to abiotic stresses (e.g. salinity, water stress) and biotic stresses (e.g. root diseases) [10,11,13,16–18]. However, biostimulants are different from conventional crop inputs in two respects. First, biostimulants affect different mechanisms of plants, regardless of their nutrient content, compared with fertilizers. Secondly, biostimulants could have an impact on the vigor of a plant without having a direct effect on pests and diseases. Therefore, they differ from crop-protection products [14]. Du Jardin [15] grouped the biostimulants into seven categories: (1) humic and fulvic acids, (2) protein hydrolysates and other N-containing compounds, (3) seaweed extracts and botanicals, (4) chitosan and other biopolymers, (5) inorganic compounds, (6) beneficial fungi, and (7) beneficial bacteria. The application form of biostimulants is important in order to have an optimal effect. Most biostimulants are applied in soil or growing media as powder, granules or a drench solution via the irrigation system [19,20]. Biostimulants can also be used as a seed treatment [21] or for foliar spray applications [22]. The interest in biostimulants is expanding worldwide as they constitute promising alternatives to unsustainable approaches. Indeed, the global biostimulant market was forecasted to reach 2.19 billion by 2018 [23] and 3 billion by 2020 [9], with an annual growth rate of 12.5% from 2019 to 2024 [12].

Although several reviews and articles [9,15,23–26] on biostimulants have been published in recent years, most of the studies were focused on the improvement of plant resilience [27–29], plant development [30] and nutrient use efficiency [31] of conventional growing crops. However, little is known about the benefits of adding biostimulants under organic farming, as organic amendments already constitute a source of beneficial fungi and bacteria, humic acids, as well as organic (e.g. amino acids, chitin) and inorganic components (e.g. Si). Based on recent studies, we have selected the most promising biostimulants for strawberries that can improve plant development, crop productivity and berry quality under conventional and organic growing conditions. Our main objective was to identify the benefits of the selected biostimulants in terms of plant growth and development, yield and fruit quality of conventional and organic strawberries grown under a protected environment. The specific objectives were (1) to compare the agronomic performance of plants grown with and without biostimulants in organic and conventional growing systems, (2) to study the effects of biostimulants on berry quality in terms of fruit size, taste, and the nutritional value, and (3) to evaluate the impact of some biostimulants and growing systems on the soil biological activity.

## 2. Materials and Methods

### 2.1. Greenhouse Experiment

A greenhouse experiment was conducted in the high-performance greenhouse complex at Laval University (Lat. 46°78′ N; long. 71°28′ W). Day-neutral strawberry tray plants, *Fragaria × ananassa* Duch. Cv. Monterey provided by FIO Inc. (Île d'Orléans, QC, Canada), were placed in 1.9 L pots (1 plant per pot) filled with a standard (BM4 40 NFW with wood fibre and peat) or organic (OM4 40 NFW with wood fibre, peat, and compost) growing media provided by Berger (Tourbières Berger, Saint-Modeste, QC, Canada). Plants were grown from February 5 to July 11, 2018, under natural light supplemented with HPS lamps providing a PPFD of 162 µmol m$^{-2}$ s$^{-1}$ at the plant level, for a photoperiod of 16 h (from 8 a.m. to 24 p.m.), with a $CO_2$ concentration of 400–500 µL L$^{-1}$, day/night temperature of 18/13 ± 0.8 °C, and a vapor pressure deficit of 1.27 kPa. Plants were irrigated with liquid organic (0.3% of Nature's Source 3-1-1 and 0.00035% of potassium silicate) or synthetic commercial fertilizers (77 mg N L$^{-1}$, 55 mg P L$^{-1}$, 164 mg K L$^{-1}$, 55 mg Ca L$^{-1}$, 20 mg Mg L$^{-1}$, 1.4 mg Fe L$^{-1}$, 1.0 mg Mn L$^{-1}$, 0.4 mg Zn L$^{-1}$, 0.3 mg Cu L$^{-1}$, 0.2 mg B L$^{-1}$, 0.01 mg Mo L$^{-1}$) [32]. The plants were fertigated twice a day at vegetative growth stage and three times per day at flowering, productive and mature stages, with a duration of three minutes. The amount of nutrient solution was 360 mL day$^{-1}$. For the organic growing system, 5.5 g of poultry manure pellets (5-3-2; Acti-sol Inc., Notre-Dame-du-Bon-Conseil, QC, Canada) was applied to the organic growing plants twice a month. Bumblebees (Biobest®, Leamington, ON, Canada), as natural pollinators were used to improve flower pollination.

### 2.2. High-Tunnel Experiment

The high-tunnel trial was performed at the farm Les Fraises de l'Ile d'Orleans Inc. (Lat. 46°51.789285′ N; long. 71°1.57608′ W) from 10 May to 2 October 2018. The tray plants (*Fragaria × ananassa* Duch. Cv. Monterey provided by FIO Inc., Île d'Orléans, QC, Canada) were cultivated in high tunnels of 4.8 m in height, 91.4 m in length and 8.4 m in width per bay covered with a simple polyethylene plastic film. The sides of the tunnel were opened to allow ventilation. Before transplanting, raised demarcated beds (40 cm in height, 25 cm in width and 15 cm in depth) were prepared and covered with tight black plastic film. A drainpipe was laid at the bottom of the bed, and then filled with peat-based growing medium (BM4 40 NF Wood 25 with wood fibres and peat, Tourbières Berger, Saint-Modeste, QC, Canada), providing 2.28 L per plant. Strawberries were planted at a distance of 20 cm on a double row in zigzag form (staggered) with a plant density of 56,250 plants per ha (10 plants per linear meter) and 60 plants per experimental unit. A drip irrigation system ensured the fertigation of the plants. The irrigation pipes were placed in the middle of each row at a rate of 9.8 holes/linear meter. According to plant development, plants were irrigated once or twice a day with synthetic commercial fertilizers with a volume of 700 mL per irrigation per plant. Like the greenhouse trial, bumblebees were used to improve the pollination of the plants.

### 2.3. Treatments

In the greenhouse, a set of 13 treatments were compared under conventional and organic growing systems: 1-conventional control without any biostimulants, 2-seaweed extract (Acadian Sea plants Lte, Dartmouth, NS, Canada), 3-*Trichoderma harzianum* strain T22, 4-*Rhizoglomus irregulare* (Tourbières Berger, Saint-Modeste, QC, Canada), 5-*Azospirillum brasilense* (free nitrogen fixator and denitrification), *Gluconacetobacter diazotrophicus* (endosymbiotic nitrogen scavenger), and *Bacillus amyloliquefaciens* (phosphate and potassium solubilizing bacteria) (Tourbières Berger, Saint-Modeste, QC, Canada), 6-mixture of mycorrhiza and nitrogen-fixing endosymbiosis bacteria (treatments 4 and 5), and 7-citric acid-based formulation (Fungout®, pH = 6.2; AEF GLOBAL Inc., Lévis, QC, Canada). For the organic growing system, the biostimulant treatments were: 8-organic control without

any biostimulants, 9-seaweed extract, 10-*Rhizoglomus irregulare*, 11-*Azospirillum brasilense*, *Gluconacetobacter diazotrophicus*, and *Bacillus amyloliquefaciens*, 12-mixture of mycorrhiza and nitrogen-fixing endosymbiosis bacteria (treatments 10 and 11), and 13-citric acid-based formulation. Seaweed extract was applied to the substrate twice a month during the experiment at a concentration of 0.4%. Citric acid was sprayed on the aerial part of the plants twice a month at a concentration of 1.25% by using a hand sprayer on the leaves and green fruits until runoff. Berger-based biostimulants (undisclosed formulation) were added to the growing media before plantation.

In the high-tunnel, five treatments were used: 1-control without any biostimulants, 2-*Rhizoglomus irregulare*, 3-*Azospirillum brasilense*, *Gluconacetobacter diazotrophicus* and *Bacillus amyloliquefaciens*, 4-mixture of mycorrhiza and nitrogen-fixing endosymbiosis bacteria (treatments 2 and 3), and 5-citric acid-based formulation (Fungout®, pH = 6.2, AEF GLOBAL Inc., Lévis, QC, Canada). Citric acid was sprayed on the aerial part of plants twice a month with a concentration of 1.25%. Mycorrhiza and bacteria were added to the growing media before plantation (nondisclosure formulation).

*2.4. Measured Parameters*

2.4.1. Soil Biological Activity

Fresh soil composite samples were prepared to determine the soil biological activity based on the total microbial population. For each experiment and experimental unit, a total of 30 g of soil was sampled (3 subsamples from each experimental unit) at 2-10 cm deep using a trowel. Sampling was performed in the morning before the first irrigation. There were, therefore, 4 and 5 replicates for each treatment in the high tunnel and greenhouse, respectively. Soil samples were stored at 4 °C for a maximum of one to two days before analysis.

The biological activity was determined by the hydrolysis of fluorescein diacetate (FDA) described by Adam and Duncan [33], which measured the enzymatic activity produced by several microorganism enzymes. Briefly, 2 g of fresh soil was added to the tubes containing 30 mL of 60 mM potassium phosphate buffer with pH 7.6. The enzymatic reaction was initiated by adding 600 μL mL$^{-1}$ of a 1000 μg fluorescein diacetate solution to each sample tube. Then, the tubes were incubated and shacked at 200 rpm at 30 °C for 20 min. After this step, tubes were centrifuged at 4500 rpm for five minutes. A standard curve (0, 1, 3, 5, and 10 μL mL$^{-1}$) was produced by diluting 0.5 mL of the solution of 2000 μg mL$^{-1}$ fluorescein in 49.5 mL of 60 mM potassium phosphate buffer. The hydrolysis of the FDA was measured 40 min after the beginning of the reaction at 490 nm using a spectrophotometer (BioTek Instruments Inc., Epoch 2 Microplate reader, Winooski, VT, USA). A more pronounced yellow color indicated higher enzymatic activity and, consequently, higher microbial activity of the sample.

2.4.2. Chlorophyll Fluorescence

Chlorophyll fluorescence (ChlF) analysis was performed using a Handy PEA fluorimeter (Handy Plant efficiency analyzer, Hansatech Instruments Ltd., King's Lynn, UK). The chlorophyll fluorescence was measured three times during the season: (1) at the flowering, (2) productive, and (3) mature stages. The measured leaves were dark-adapted for 20 min by attaching light-exclusion clips to the leaf's surface, avoiding the central vein, while the plants were in the light. The Fv (variable fluorescence), Fm (maximum fluorescence), maximum Fv/Fm ratio (maximum quantum efficiency of photosystem II), and performance index (indicator of sample vitality) parameters were recorded for one second with 3000 μmol m$^{-2}$ s$^{-1}$ PPFD (photosynthetic photon flux density). For each experimental unit, three plants were randomly selected, and measurements were performed in the morning, one hour after irrigation (3 plants with one leaf reading per plant; 15 plants per treatment for the greenhouse experiment and 12 plants per treatment for the high tunnel experiment).

The parameters were calculated according to the equations described by Strasser et al. [34].

$$F_v / F_m = (F_M - F_0) / F_M \tag{1}$$

$$PI = \frac{1 - (F_0 / F_M)}{M_0 / V_J} \times \frac{F_M - F_0}{F_0} \times \frac{1 - V_J}{V_J} \tag{2}$$

where $F_0$ = fluorescence intensity at 50 µs, $F_J$ = fluorescence intensity at the J step (at 2 ms), $F_M$ = maximal fluorescence intensity, $V_J$ = relative variable fluorescence at 2 ms calculated as $V_J = (F_J - F_0)/(F_M - F_0)$, $M_0$ = initial slop of fluorescence kinetics, which can be derived from the Equation (3):

$$M_0 = 4 \times (F_{300\mu s} - F_0) / (F_M - F_0) \tag{3}$$

### 2.4.3. Chlorophyll Content and Photosynthetic Parameters

Measurements were performed in the morning, by starting one hour after irrigation. The leaf chlorophyll content was measured by using a chlorophyll meter (SPAD-502, Minolta corporation, Ltd., Chuo-ku, Osaka, Japan) on the same leaf where the chlorophyll fluorescence was measured. The chlorophyll content was determined by the average of three readings per leaf for a total of nine measurements per experimental unit (three plants with three leaf readings per plant). Leaf photosynthesis light–response curves were performed on well-exposed mature leaves of one plant per experimental unit for both experiments by using a portable gas-exchange system, model LI-6400XT (LI-COR Inc., Lincoln, NE, USA). Briefly, the measurement system was set at 1800 µmol m$^{-2}$ s$^{-1}$ photosynthetically active radiation (PAR), an air temperature of 24 °C, a vapor pressure deficit (VPD) of 1.3 kPa, a leaf chamber $CO_2$ concentration of 450 µmol mol$^{-1}$, and a flow rate of 350 µmol s$^{-1}$. After around 15 min of acclimation, the light intensity was varied from high to low PAR (1800, 1500, 1200, 900, 700, 550, 375, 275, 200, 150, 100, 75, 50, 20, and 1 µmol photons m$^{-2}$ s$^{-1}$), and the gas-exchange parameters were recorded for each light level. Then, photosynthetic parameters, such as the dark respiration rate (Rd), quantum efficiency (Φ), and maximum rate of photosynthesis (Amax) were extracted from the curves as described by Hansen et al. [35].

### 2.4.4. Non-Destructive Growth Parameters

Plant growth was measured monthly on three random samples of strawberry plants per experimental unit at the flowering, productive and mature stages. The measured parameters included the number of leaves, number of flowering fruit stalks, number and the diameter of crowns by using a digital caliper (Neoteck 6 inches, Kowloon, HK, Hong Kong).

### 2.4.5. Foliar Mineral Analysis

Leaf sampling was performed for both experiments from each experimental unit to determine their mineral content (N, P, K, Ca, and Mg). Mineral analysis was measured twice in the season at the flowering and productive stages. Three fully developed leaves from three plants per experimental unit were sampled. The concentrations of nutrients were determined based on the percent or ppm of dry matter. The samples were placed in well-identified paper bags and dried at 60 °C for 48 h. The total nitrogen was determined by a CNS-Leco 2000 analyzer. The available elements were extracted according to the Mehlich III method [36], while the sparingly soluble forms were extracted with ammonium acid oxalate. The total phosphorus (Pt) and inorganic phosphorus (Pi) were extracted with sulfuric acid (0.5 M $H_2SO_4$) and calcined at 480 °C prior to extraction for the analysis of total phosphorus [36]. Organic phosphorus was calculated by the difference

between Pt and Pi. The determination was carried out by atomic absorption spectrophotometry for Ca and Mg, and by emission in the flame for K. Phosphorus was measured colorimetrically at 660 nm by the blue method for Pi and Pt [37] and by the PB-PC method for Mehlich III extraction [36].

### 2.4.6. Yield

The fruit yield was evaluated once or twice a week for the greenhouse experiment, and three times a week for the high tunnel experiment. At each harvest, fruit classification was performed according to the shape and fruit size (calibre). The fruits were then classified into two groups: marketable and unmarketable fruits. For each treatment, the number and weight of the fruits were recorded. Some fruit was considered unmarketable when smaller than 5 g and 1.90 cm, along with exhibiting signs of disease and poor pollination.

### 2.4.7. Total Sugar Level (°Brix)

The soluble sugar content (SSC) or °Brix is a sweetness measurement, measured by using a refractometer (Atago PAL-1 (3810), Tokyo, Japan). °Brix was evaluated twice a month in both experiments (four measurements for the greenhouse and six measurements for the high tunnel). Note that the fruit yield occurred faster in the high tunnel. Therefore, we had four additional weeks of harvest in the high tunnel compared with the greenhouse experiments, resulting two extra samples.

Briefly, fully ripe fruits were harvested on the day of measurement. Three ripe fruits from each experimental unit were selected for the °Brix measurement. Fruits were crushed using a blender or garlic press, and then the pulp and seeds were removed using filter paper. A few drops of the sample juice were placed on a refractometer using a plastic pipette to record the %SSC or °Brix value. Between each reading, the refractometer was cleaned and calibrated to 0% SSC using distilled water.

### 2.4.8. Total Phenolic Content (TPC)

Ten fruit samples were collected at optimum maturity and stored at -20 °C until analysis. The total phenolic content was evaluated twice a month in both experiments (four measurements for the greenhouse and six for the high tunnel). The TPC was measured according to Singleton and Rossi [38] using the Folin–Ciocalteu (F-C) reagent. For each sample, 0.3 g of freeze-dried powdered strawberries was mixed with 20 mL of 80% methanol, placed in an ultrasonic bath at 37 °C for 20 min, and centrifuged at 4000 rpm for 4 min. The extraction was repeated three times. After diluting the liquid extract, 20 μL of water (white extracts), the sample, and the standard were mixed with 100 μL of Folin–Ciocalteu reagent to conduct the reaction. The processing time was 1–8 min. Then, an amount of 80 μL of the 7.5% sodium carbonate solution was added ($Na_2CO_3$) to a 2 mL vial and mixed well. After 45 min, absorption was measured at 765 nm using a spectrophotometer (BioTek Instruments Inc., Winooski, VT, USA). The results were expressed in mg of gallic acid equivalent (GAE) per mL (mg GAE mL$^{-1}$) considering the sample dilution. The analyses were performed in triplicate.

### 2.4.9. Total Anthocyanin Content (TAC)

Ten ripe fruit samples per experimental unit were collected monthly (five measurements per season) and stored at −20 °C until anthocyanin analysis. The anthocyanin content of the fruits was determined by the pH differential method developed by AOAC International and approved by Lee et al. [39]. The freeze-dried powder of the berries was extracted using methanol: water: acetic acid (85: 15: 0.5 *v/v*, MeOH/H$_2$O/AcOH) as previously reported by Wu and Prior [40]. In brief, 0.3 g of the sample powder was added to 5 mL of the acidic methanol solvent and mixed well for 30 s. Then, the tubes were placed in an ultrasonic bath for 15 min. The supernatant was transferred to another tube following centrifugation for 5 min at 4000 rpm. Then, the test solution was prepared using pH 1.0

and 4.5 buffers to determine an appropriate dilution factor. After diluting the extracts, blanks were produced with pH 1.0 and 4.5 buffers. Amounts of 0.5 mL of diluted extract and 2.5 mL of the buffers were added to the 4 mL cuvettes. The solution was mixed well and left to stand for 30 min at room temperature. The samples were measured under the absorbance of 510 nm and 700 nm using a spectrophotometer (BioTek Instruments Inc., Winooski, VT, USA). The results were expressed as a concentration of pelargonidin-3-Oglycoside (mg 100 mL$^{-1}$). The difference in absorbance between the two samples was calculated using the following equation:

$$\text{Absorbance} = \left[A_{510nm}(pH1.0) - A_{700nm}(pH1.0)\right] - \left[A_{510nm}(pH4.5) - A_{700nm}(pH4.5)\right] \quad (4)$$

The concentration of the anthocyanin was calculated by cyanidin-3-glucoside equivalents as follows:

$$\% \, ^{W}/_{W} = \frac{A}{\varepsilon \times L} \times MW \times DF \times \frac{V}{Wt} \times 100 \quad (5)$$

where A = absorbance; $\varepsilon$ = 26,900 molar extinction coefficient, in L mol$^{-1}$ cm$^{-1}$, for cyd-3-glu; L = pathlength in cm; MW (molecular weight) = 449.2 g/mol for cyanidin-3-glucoside (cyd-3-glu); DF = dilution factor established in D; V = final volume of the solvent; Wt = weight of the sample.

### 2.5. Statistical Analysis

In the statistical model, the biostimulant factor had 18 modalities with treatments nested in the systems (greenhouse or tunnel). For each system, the experimental design was randomized in complete blocks. The block, date (when appropriate) and systems were considered random factors, while treatment was a fixed factor. Data normality was checked using the Shapiro–Wilk statistic and homogeneity of variance was assessed visually by examining the graphic distribution of residuals. The MIXED procedure was used with a repeated statement and a covariance structure that minimized the Akaike criterion for data with several dates of measurement. The GROUP statement and LOG transformation were used to achieve data homogeneity if necessary. Pairwise comparisons were made using protected Fisher's LSD. A contrast statement was added to compare the effects of the treatments used in organic management and treatments used in conventional management on all variables overall. Principal component analysis (PCA) was conducted on the leaf mineral concentration, physiological, yield, quality, and soil activity parameters. Spearman correlations were calculated to find links between variables without the influence of extreme data measurements. All data were analyzed by a two-way model of the analysis of variance (ANOVA) by using the MIXED procedure in SAS software (version 9.4, SAS Institute Inc., Cary, NC, USA) with significance determined at $p \leq 0.05$.

## 3. Results

### 3.1. Microbial Activity

The statistical analysis showed that for all growing systems (CG, OG, and CH) biostimulant treatments did not influence the microbial soil activity compared with its respective control (Figure 1). However, the microbial activity of soils, expressed by the hydrolysis of the fluorescence diacetate (FDA), was influenced by the organic growing system at $p < 0.001$. The soils of the organic greenhouse growing system (OG) showed a significant increase (+74%) in microbial activity compared with the same treatments in the conventional greenhouse part at $p < 0.01$ (CG). Specifically, the soil microbial activity significantly increased in the soils treated with a combination of mycorrhizae and bacteria (MYC + BACT; +98%), followed by citric acid (CITRIC; +90%), seaweed extract (SEAWEED; +74%), and mycorrhizae (MYC; +45%) in organic soil compared with the same treatments in the conventional system. In the conventional greenhouse, however, the microbial activity of the soil treated with MYC was significantly higher (+194%) compared with the conventional high tunnel.

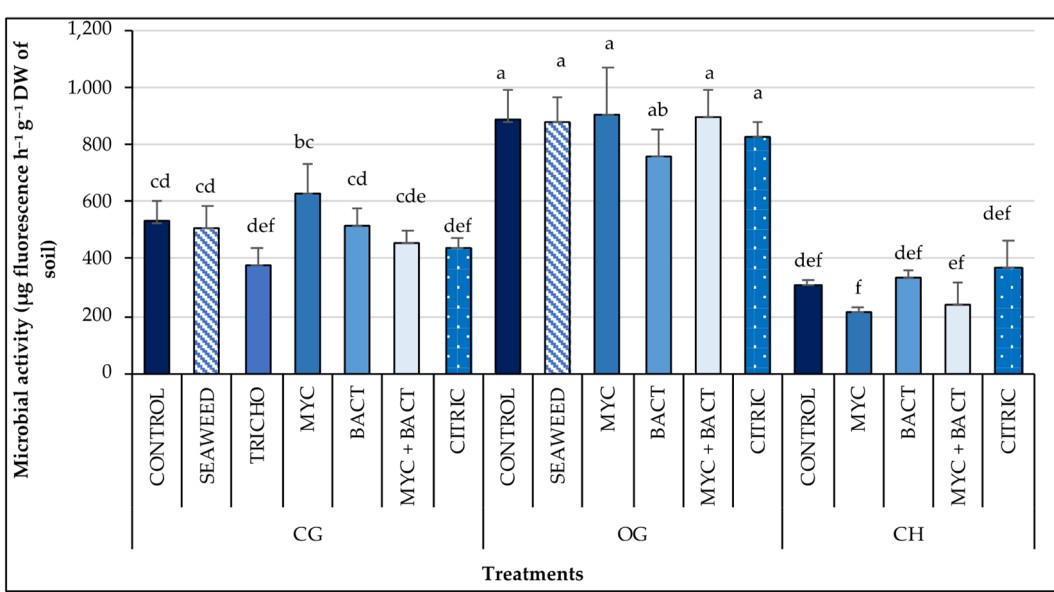

**Figure 1.** Influence of the studied biostimulants and growing systems on the microbial activity of the soil during winter and summer 2018. Means followed by the same letter are not significantly different ($p \leq 0.05$). Microbial activity is expressed in µg fluorescence $h^{-1}$ $g^{-1}$ dry weight of soil. CG: conventional greenhouse; OG: organic greenhouse; CH: conventional high tunnel; CONTROL: without biostimulant; SEAWEED: seaweed extract; TRICHO: *Trichoderma*, MYC: mycorrhiza (*Rhizoglomus irregulare*); BACT: three bacteria (*Azospirillum brasilense*, *Gluconacetobacter diazotrophicus*, and *Bacillus amyloliquefaciens*); MYC + BACT: a combination of treatments MYC and BACT; CITRIC: Citric acid.

### *3.2. Photosynthetic Parameters and Leaf Chlorophyll Content*

The chlorophyll fluorescence parameters and leaf chlorophyll content (SPAD value) are shown in Table 1. For all measured parameters, the results showed no significant differences between the biostimulant treatments and their respective control, except for the maximum quantum efficiency of photosystem II expressed by Fv/Fm of MYC in the organic greenhouse (OG), which was slightly reduced. Plants treated with citric acid (CITRIC) in the OG showed a slight, but significant increase ($p < 0.001$) in Fv/Fm compared with the same treatment in the conventional part (CG). We also observed that the biostimulant treatments in the conventional high tunnel (CH) induced higher values of Fv/Fm, P Index and leaf chlorophyll content compared with their respective CG treatments ($p < 0.001$). Regardless of the biostimulant treatments, greenhouse plants grown organically had a 11% higher performance index (PI) compared with CG grown plants, while PI of CH grown plants was 33% higher than CG. The leaf chlorophyll contents of CG and OG were similar, while a slight, but significant increase was observed for CH compared with CG.

No significant difference was observed between the biostimulant treatments for the leaf photosynthesis light–response curves measured twice during the experiments and their related parameters: dark respiration rate (Rd) of 0.592 µmol $CO_2$ $m^{-2}$ $s^{-1}$ for CG and OG and 1.342 µmol $CO_2$ $m^{-2}$ $s^{-1}$ for CH; quantum efficiency (Φ) of 0.074 mol $CO_2$ fixed $mol^{-1}$ absorbed photons for CG and OG and 0.075 mol $CO_2$ fixed $mol^{-1}$ absorbed photons for CH; maximum rate of photosynthesis (Amax) of 17.39 µmol $CO_2$ $m^{-2}$ $s^{-1}$ for CG and OG and 16.70 µmol $CO_2$ $m^{-2}$ $s^{-1}$ for CH (data not shown). However, Amax of CG grown plants were higher than OG (18.32 vs 16.45 µmol $CO_2$ $m^{-2}$ $s^{-1}$, $p = 0.008$).

**Table 1.** Influence of biostimulant treatments on chlorophyll fluorescence parameters, Fv/Fm and performance index (PI), and leaf chlorophyll content (SPAD) of strawberries grown conventionally and organically under a greenhouse and high tunnel during winter and summer 2018.

| Treatments | | Fv/Fm | PI | SPAD |
|---|---|---|---|---|
| CG [z] | CONTROL | 0.805 bcd [x] | 2.8 fgh | 37.3 def |
| | SEAWEED | 0.803 bcde | 2.7 fgh | 37.0 ef |
| | TRICHO | 0.804 bcd | 3.1 defg | 37.1 ef |
| | MYC | 0.798 e * | 2.6 h | 36.7 f |
| | BACT | 0.802 cde | 2.6 gh | 37.2 def |
| | MYC + BACT | 0.801 de | 2.7 fgh | 37.4 def |
| | CITRIC | 0.801 de | 2.6 fgh | 36.8 ef |
| OG | CONTROL | 0.803 bcde | 3.0 efgh | 38.1 cde |
| | SEAWEED | 0.808 bc | 3.1 defg | 37.3 def |
| | MYC | 0.803 bcde | 2.9 efgh | 38.2 cde |
| | BACT | 0.802 bcde | 2.8 fgh | 37.7 cdef |
| | MYC + BACT | 0.803 bcde | 3.0 efgh | 37.1 ef |
| | CITRIC | 0.808 b | 3.1 cdef | 38.1 cde |
| CH | CONTROL | 0.821 a | 3.5 abcd | 39.9 ab |
| | MYC | 0.824 a | 3.9 a | 40.3 a |
| | BACT | 0.824 a | 3.7 ab | 39.1 abc |
| | MYC + BACT | 0.821 a | 3.5 abc | 40.6 a |
| | CITRIC | 0.820 a | 3.3 bcde | 38.7 bcd |
| OG vs. CG | OG | 0.804 a | 3.0 a | 37.8 |
| | CG | 0.802 b | 2.7 b | 37.1 |
| CG vs. CH | CH | 0.822 a | 3.6 a | 39.7 a |
| | CG | 0.802 b | 2.7 b | 37.1 b |
| *p* values | | | | |
| Biostimulant (B) | | <0.001 | <0.001 | <0.001 |
| OG vs. CG | | 0.042 | 0.007 | 0.012 |
| CG vs. CH | | <0.001 | <0.001 | <0.001 |

[z] CG: conventional greenhouse; OG: organic greenhouse; CH: conventional high tunnel; CONTROL: without biostimulant; SEAWEED: seaweed extract; TRICHO: *Trichoderma*, MYC: mycorrhiza *Rhizoglomus irregulare*; BACT: three bacteria—*Azospirillum brasilense*, *Gluconacetobacter diazotrophicus*, and *Bacillus amyloliquefaciens*; MYC + BACT: a combination of treatments MYC and BACT; CITRIC: Citric acid. [x] means of the same column with different letters are significantly different at $p < 0.05$. * Treatments are different from their respective control.

### 3.3. Growth Parameters

Our results showed that non-destructive growth parameters, such as the number of leaves ($p = 0.04$), number of flowering stalks ($p < 0.001$), number of crowns ($p < 0.001$) and the diameter of crowns ($p < 0.001$) were impacted by the application of biostimulants (Table 2). Although, the biostimulant treatments did not improve plant growth cultivated in the greenhouse compared with their control treatments, BACT, MYC + BACT and CITRIC treatments increased the number of flowering stalks of plants grown in CH by 20%, 25% and 32%, respectively, compared with their control treatment, while BACT increased the diameter of crowns by 12%. At the end of the experiments, all biostimulants increased in average the dry shoot plant biomass of strawberries grown under CH by 55% compared with the control (28.72 vs 18.47 g plant[-1]), while no positive impact was observed for CG and OG grown plants (average of 33.13 and 26.81 g plant[-1] for CG and OG, respectively) (data not shown). Regarding the growing systems, all non-destructive growth parameters were higher ($p < 0.001$) for plants cultivated in the high tunnel (CH) than in the greenhouse (CG and OG). Indeed, the number of leaves, number of flowering stalks, number of crowns, and the diameter of crowns were significantly higher for plants grown in CH and treated with MYC (+33%, +48%, +17%, and 3+1%), BACT (+42%, +73%, +31%, and +37%),

MYC + BACT (+46%, +102%, +37%, and +29%), and CITRIC (+24%, +64%, +5%n.s., and +20%) compared with the CG.

**Table 2.** Influence of biostimulant treatments on non-destructive growth parameters of strawberries grown conventionally and organically under a greenhouse and high tunnel during winter and summer 2018.

| Treatments | | Number of Leaves | Number of Flowering Stalks | Number of Crowns | Diameter of Crowns (mm) |
|---|---|---|---|---|---|
| CG [z] | CONTROL | 15.9 cd [x] | 4.9 efg | 3.5 cdefg | 38.1 defg |
| | SEAWEED | 14.5 d | 4.7 efg | 3.2 fg | 36.5 defg |
| | TRICHO | 16.3 cd | 5.1 ef | 3.5 cdefg | 39.4 cde |
| | MYC | 15.4 cd | 4.8 efg | 3.3 efg | 36.9 defg |
| | BACT | 15.3 cd | 4.5 efg | 3.3 efg | 36.1 efg |
| | MYC + BACT | 14.1 d | 4.0 g | 3.2 efg | 34.4 fg |
| | CITRIC | 16.4 cd | 5.2 def | 3.7 bcde | 38.2 def |
| OG | CONTROL | 15.9 cd | 4.8 efg | 3.4 defg | 40.2 cd |
| | SEAWEED | 14.7 d | 4.3 fg | 3.4 efg | 34.2 g * |
| | MYC | 15.2 cd | 4.0 g | 3.2 g | 37.9 defg |
| | BACT | 15.2 cd | 5.1 ef | 3.3 efg | 37.0 defg |
| | MYC + BACT | 15.5 cd | 5.2 ef | 3.4 defg | 36.9 defg |
| | CITRIC | 17.3 bc | 5.5 de | 3.6 bcdef | 39.4 cde |
| CH | CONTROL | 21.1 a | 6.5 cd | 4.1 ab | 44.2 bc |
| | MYC | 20.4 ab | 7.1 bc | 3.9 abcd | 48.1 ab |
| | BACT | 21.8 a | 7.8 ab * | 4.3 a | 49.5 a * |
| | MYC + BACT | 20.6 ab | 8.1 ab * | 4.4 a | 44.5 bc |
| | CITRIC | 20.4 ab | 8.6 a * | 3.9 abc | 45.8 ab |
| OG vs. CG | OG | 15.6 | 4.8 | 3.4 | 37.6 |
| | CG | 15.4 | 4.7 | 3.4 | 37.1 |
| CG vs. CH | CH | 20.8 a | 7.6 a | 4.1 a | 46.4 a |
| | CG | 15.4 b | 4.7 b | 3.4 b | 37.1 b |
| *p* values | | | | | |
| Biostimulant (B) | | 0.040 | <0.001 | <0.001 | <0.001 |
| OG vs. CG | | 0.644 | 0.697 | 0.900 | 0.548 |
| CG vs. CH | | <0.001 | <0.001 | <0.001 | <0.001 |

[z] CG: conventional greenhouse; OG: organic greenhouse; CH: conventional high tunnel; CONTROL: without biostimulant; SEAWEED: seaweed extract; TRICHO: *Trichoderma*, MYC: mycorrhiza *Rhizoglomus irregulare*; BACT: three bacteria—*Azospirillum brasilense*, *Gluconacetobacter diazotrophicus*, and *Bacillus amyloliquefaciens*; MYC + BACT: a combination of treatments MYC and BACT; CITRIC: Citric acid. [x] means of the same column with different letters are significantly different at *p* < 0.05. * Treatments are different from their respective controls.

### 3.4. Foliar Mineral Content

The biostimulant treatments and growing systems significantly affected the leaf mineral concentration as shown in Table 3.

**Table 3.** Influence of biostimulant treatments on leaf mineral concentrations (% of the leaf dry weight) of strawberry plants grown conventionally and organically under a greenhouse and high tunnel during winter and summer 2018.

| Treatments | | N (%) | P (%) | K (%) | Ca (%) | Mg (%) |
|---|---|---|---|---|---|---|
| CG [z] | CONTROL | 1.83 fg [x] | 0.471 a | 1.41 | 0.634 cdef | 0.186 |
| | SEAWEED | 1.93 efg | 0.466 a | 1.36 | 0.670 bcdef | 0.198 |
| | TRICHO | 1.98 bcdef | 0.494 a | 1.42 | 0.678 bcdef | 0.210 |
| | MYC | 2.02 abcdef | 0.486 a | 1.41 | 0.740 abcde | 0.209 |
| | BACT | 1.89 fg | 0.477 a | 1.32 | 0.685 bcdef | 0.199 |
| | MYC + BACT | 1.95 efg | 0.457 ab | 1.35 | 0.571 efg | 0.182 |
| | CITRIC | 1.86 fg | 0.460 ab | 1.42 | 0.809 ab * | 0.226 |

| | | | | | | |
|---|---|---|---|---|---|---|
| OG | CONTROL | 2.19 abc | 0.396 bc | 1.45 | 0.532 fg | 0.212 |
| | SEAWEED | 2.15abcde | 0.389 bc | 1.47 | 0.434 g | 0.200 |
| | MYC | 2.00 abcdef | 0.363 cd | 1.38 | 0.520 fg | 0.206 |
| | BACT | 2.21 a | 0.382 cd | 1.36 | 0.612 cdef | 0.227 |
| | MYC + BACT | 2.18 abcd | 0.365 cd | 1.39 | 0.593 defg | 0.235 |
| | CITRIC | 1.96 defg * | 0.390 bc | 1.43 | 0.591 defg | 0.210 |
| CH | CONTROL | 1.86 g | 0.276de | 1.38 | 0.903 a | 0.196 |
| | MYC | 1.94 abcdef * | 0.242 e | 1.43 | 0.851 ab | 0.188 |
| | BACT | 1.99 abcdef * | 0.296 cde | 1.26 | 0.889 a | 0.220 |
| | MYC + BACT | 2.09 abcdef * | 0.236 e | 1.37 | 0.775 abcd | 0.189 |
| | CITRIC | 2.11 abcdef * | 0.233 e | 1.41 | 0.634 abc | 0.221 |
| OG vs. CG | OG | 2.1 a | 0.381 b | 1.4 | 0.547 b | 0.215 |
| | CG | 1.9 b | 0.473 a | 1.4 | 0.684 a | 0.201 |
| CG vs. CH | CH | 2.0 | 0.257 b | 1.4 | 0.844 a | 0.203 |
| | CG | 1.9 | 0.473 a | 1.4 | 0.684 b | 0.201 |
| *p* values | | | | | | |
| Biostimulant (B) | | 0.008 | <0.001 | <0.001 | <0.001 | 0.281 |
| OG vs. CG | | <0.001 | <0.001 | 0.285 | <0.001 | 0.076 |
| CG vs. CH | | 0.570 | <0.001 | 0.804 | <0.001 | 0.915 |

[z] CG: conventional greenhouse; OG: organic greenhouse; CH: conventional high tunnel; CONTROL: without biostimulant; SEAWEED: seaweed extract; TRICHO: *Trichoderma*, MYC: mycorrhiza *Rhizoglomus irregulare*; BACT: three bacteria—*Azospirillum brasilense*, *Gluconacetobacter diazotrophicus*, and *Bacillus amyloliquefaciens*; MYC + BACT: a combination of treatments MYC and BACT; CITRIC: Citric acid. [x] means of the same column with different letters are significantly different at $p < 0.05$. * Treatments are different from their respective control.

Nitrogen—Biostimulant treatments did not impact the N leaf concentration of greenhouse grown strawberries (CG and OG) compared with their respective control, except for organically grown plants treated with citric acid where lower content (-10%) was observed. However, organically grown plants (OG) treated with BACT and MYC + BACT had higher N concentrations (17% and 12%) compared with conventional plants (CG) of the same treatments. Regardless of the biostimulant treatments, leaves of OG grown plants had 10% higher N content than those of CG grown plants. For conventionally plants grown under a high tunnel, all biostimulant treatments increased leaf N concentration by 9 to 13% compared with its control (Table 3).

Phosphorus—Biostimulant treatments did not significantly affect the leaf P concentration of conventionally (CG and CH) and organically (OG) grown plants compared with their respective control (Table 3). Regardless of the biostimulant treatments, conventionally greenhouse grown plants had higher P concentration (+24%) than organically grown plants, while plants grown conventionally in the greenhouse showed higher P concentration than the conventional high tunnel.

Potassium—Similarly to the leaf P concentration, biostimulant treatments did not significantly affect the leaf K concentration of plants compared to their respective control. In addition, we did not observe any difference between the three growing systems (Table 3).

Calcium—Biostimulant treatments did not significantly increase the leaf Ca concentration of conventionally (CG and CH) and organically grown plants compared with their control, except for CG grown plants treated with citric acid where higher content (28%) was observed. Regardless of the biostimulant treatments, the leaf Ca concentration of OG grown plants was 20% lower than the CG grown plants. Treatments with MYC (42%), CITRIC (37%), and SEAWEED (54%) in CG showed higher Ca concentration than their respective OG. In addition, the Ca concentration of plants grown in the CH was higher in the treatments with BACT (30%) and MYC + BACT (35%) than CG (Table 3).

Magnesium—For leaf Mg concentration, there was no significant difference between biostimulant treatments neither among the growing systems (Table 3).

### 3.5. Yield and Fruit Size

The results of the total (marketable and unmarketable) and marketable yields, and the total number of harvested fruits are presented in Figure 2. A significant difference between the biostimulant treatments ($p < 0.001$) and growing systems ($p < 0.001$) was observed for all yield parameters. Strawberry plants with the foliar application of citric acid (CITRIC) in GC produced a higher number of fruits, total yield, and marketable yield (+17% on average) compared with its respective control, while for OG grown strawberries, citric acid only increased the total number of fruits harvested per week (+16%). Regardless of the biostimulant treatments, CG grown plants produced a higher total yield (+16%) and weight of marketable fruits (+20%) than OG grown plants.

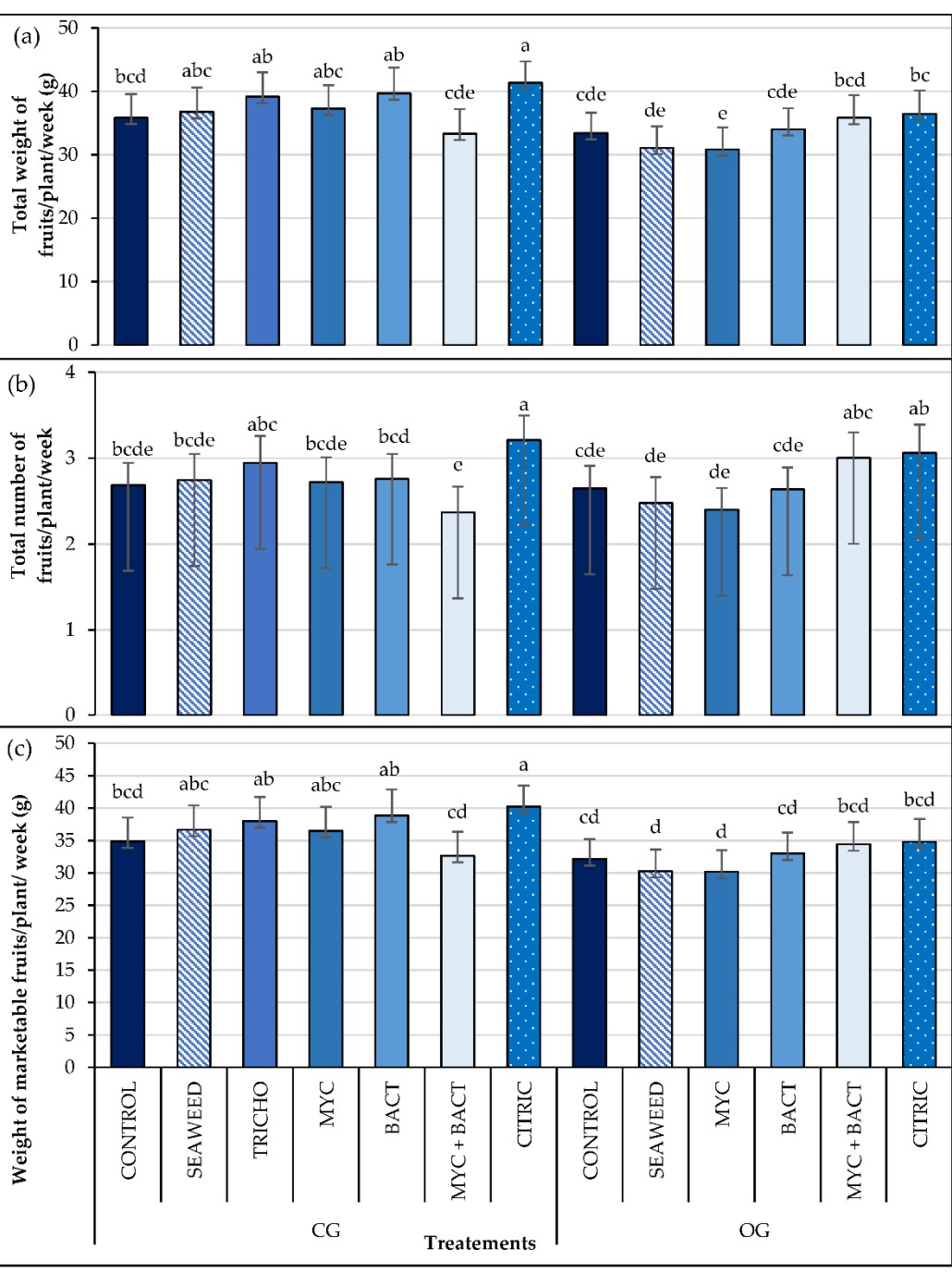

**Figure 2.** The influence of biostimulants and greenhouse growing systems on the total weight of fruits harvested per plant and per week (**a**), total number of harvested fruits per plant and per

week (**b**), and the weight of marketable fruits harvested per plant and per week (**c**) in the greenhouse during winter 2018. Means followed by the same letter are not significantly different ($p \le$ 0.05). CG: conventional greenhouse; OG: organic greenhouse; CONTROL: without biostimulant; SEAWEED: seaweed extract; TRICHO: *Trichoderma*, MYC: mycorrhiza (*Rhizoglomus irregulare*); BACT: three bacteria (*Azospirillum brasilense*, *Gluconacetobacter diazotrophicus*, and *Bacillus amyloliquefaciens*); MYC + BACT: a combination of treatments MYC and BACT; CITRIC: citric acid.

In the conventional high tunnel, MYC increased the total yield by 10% (Figure 3a), while citric acid (CITRIC) increased the number of fruits by 10% (Figure b). No significant difference was observed for the marketable yield and the other biostimulants (Figure 3).

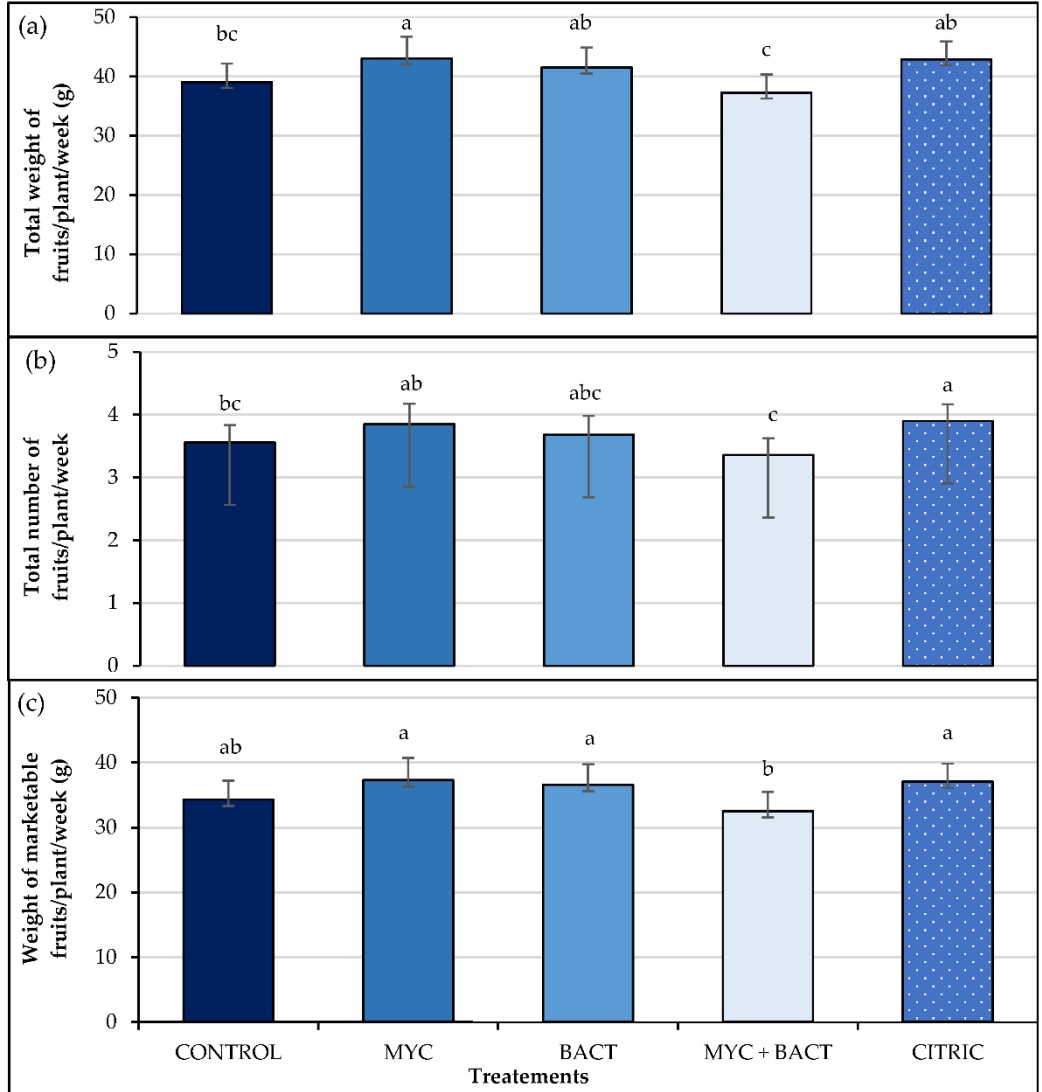

**Figure 3.** The influence of biostimulants on the total weight of fruits harvested per plant and per week (**a**), total number of fruits harvested per plant and per week (**b**), weight of marketable fruits harvested per plant and per week (**c**) for strawberries grown under high tunnel during summer 2018. Means followed by the same letter are not significantly different ($p \le 0.05$). CONTROL: without biostimulant; MYC: mycorrhiza (*Rhizoglomus irregulare*); BACT: three bacteria (*Azospirillum brasilense*, *Gluconacetobacter diazotrophicus*, and *Bacillus amyloliquefaciens*); MYC + BACT: combination of treatments MYC and BACT; CITRIC: citric acid.

With regard to fruit size, no significant difference was measured between the biostimulant treatments and their respective control (Figure 4). Regardless of the biostimulant treatments, fruits harvested under the CG showed larger fruit size compared with OG and CH ($p < 0.001$), while no difference was observed between OG and CH.

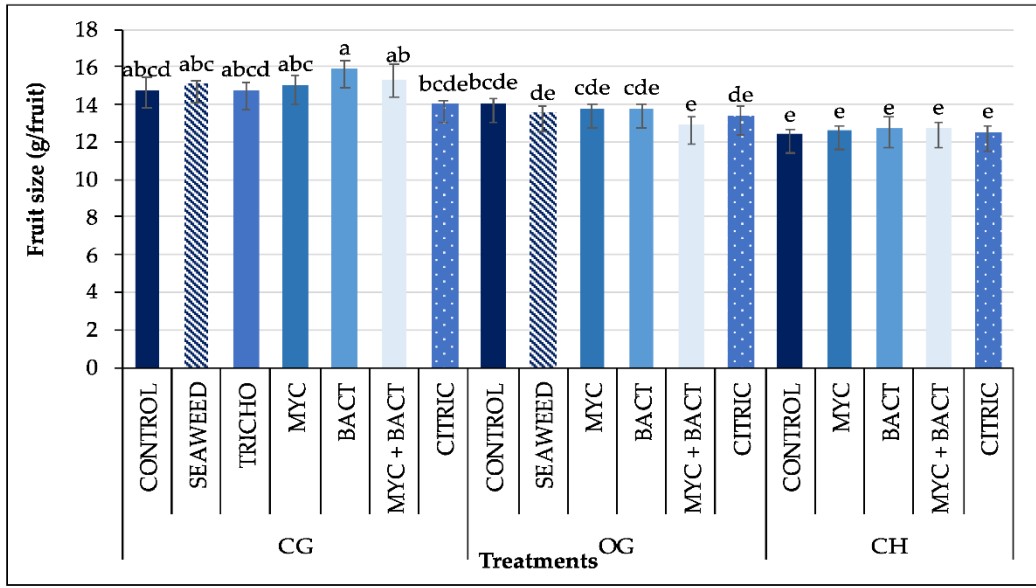

**Figure 4.** The influence of biostimulants and growing systems on fruit size of strawberries grown in winter under a greenhouse and in summer under a high tunnel. Means followed by the same letter are not significantly different ($p \leq 0.05$). CG: conventional greenhouse; OG: organic greenhouse; CH: conventional high tunnel; CONTROL: without biostimulant; SEAWEED: seaweed extract; TRICHO: *Trichoderma*, MYC: mycorrhiza (*Rhizoglomus irregulare*); BACT: three bacteria (*Azospirillum brasilense*, *Gluconacetobacter diazotrophicus*, and *Bacillus amyloliquefaciens*); MYC + BACT: a combination of treatments MYC and BACT; CITRIC: citric acid.

### 3.6. Fruit Quality

The soluble sugar content (°Brix) ($p < 0.001$), total polyphenols ($p = 0.003$), and anthocyanin content ($p < 0.001$) were significantly influenced by the application of biostimulants (Figure 5). Plants treated with *Trichoderma* (TRICHO) in the CG and the mixture of mycorrhiza and bacteria (MYC + BACT) in the organic system (OG) produced fruits with a higher °Brix (+12%) and total polyphenol content (+31% and +40%, respectively) compared with their control (Figure 5a). In the CG, all biostimulant treatments significantly increased the anthocyanin content compared with their respective control, except for bacteria (BACT) (Figure 5c). The mixture of mycorrhiza and bacteria (MYC + BACT) in the OG produced fruits with a higher anthocyanin content (+17%) compared with the control. All treatments in the CH significantly increased the content of anthocyanins compared with their control (Figure 5c), while no significant effect was observed for the °Brix and polyphenol content (Figure 5a.b).

However, regardless of the biostimulant treatments, no significant difference was observed between the organic and conventional growing systems in the greenhouse (CG vs. OG) and between CG and CH.

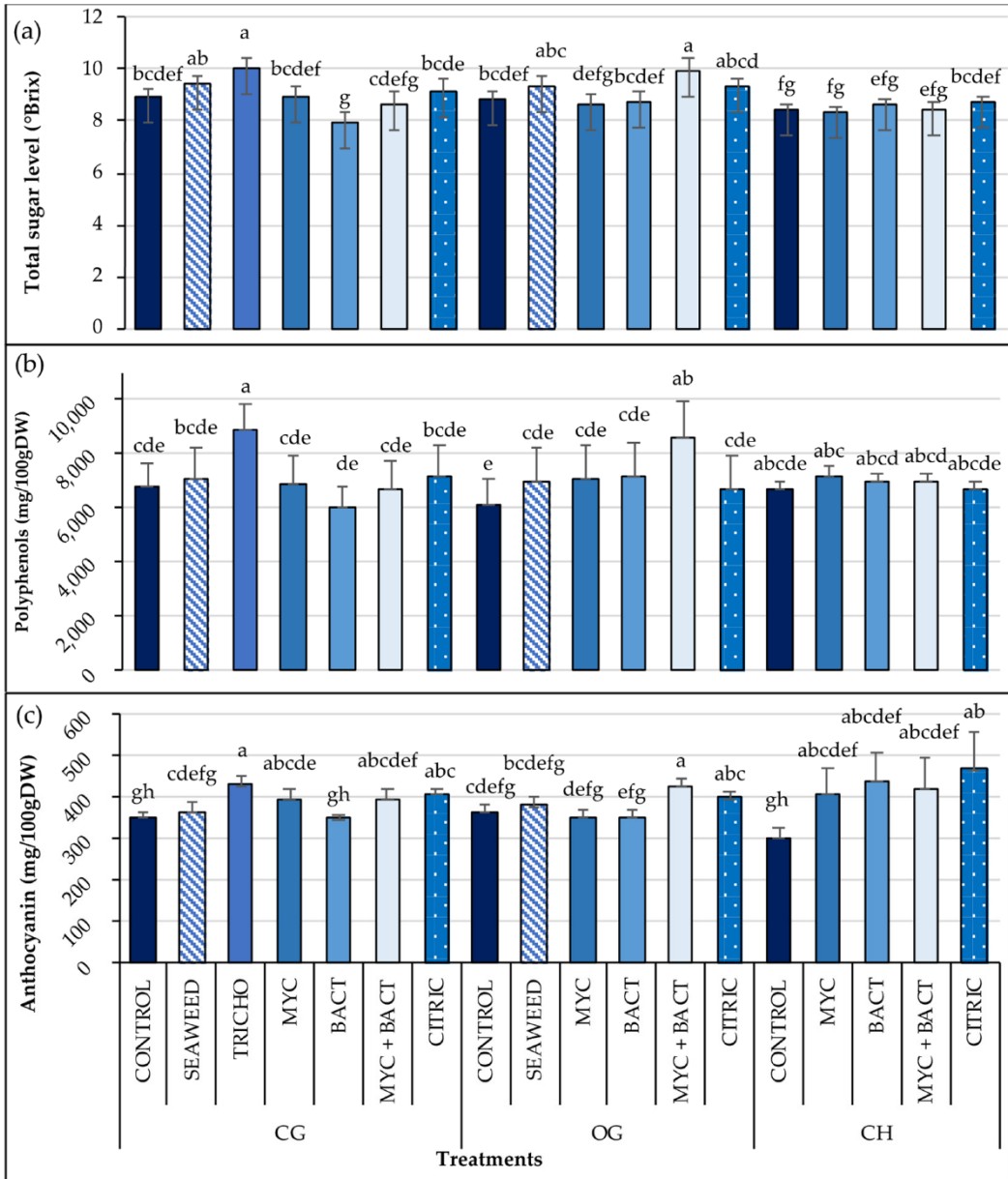

**Figure 5.** The influence of biostimulants and growing system on the total sugar level (**a**), total polyphenol content (**b**), and anthocyanin content (**c**) of berries harvested in winter and summer 2018. Means followed by the same letter are not significantly different ($p \leq 0.05$). CG: conventional greenhouse; OG: organic greenhouse; CH: conventional high tunnel; CONTROL: without biostimulant; SEAWEED: seaweed extract; TRICHO: *Trichoderma*, MYC: mycorrhiza (*Rhizoglomus irregulare*); BACT: three bacteria (*Azospirillum brasilense*, *Gluconacetobacter diazotrophicus*, and *Bacillus amyloliquefaciens*); MYC + BACT: a combination of treatments MYC and BACT; CITRIC: citric acid.

*3.7. Principal Component Analysis (PCA)*

Principal component analysis (PCA) was used to represent the relationship between some important variables and biostimulant treatments (Figure 6). PC1 and PC2 explained 81.86% of the total variance, accounting for 69.66% and 12.2%, respectively. An apparent clustering among both growing systems and growing conditions was observed. The conventional greenhouse treatments (CG 01 to CG 07) are all located in the lower-left quadrat of the figure, while the organic greenhouse treatments (OG 01 to OG 06) are all found in the upper-left quadrat. Additionally, the conventional high tunnel treatments are located in the two right quadrats (Figure 6a).

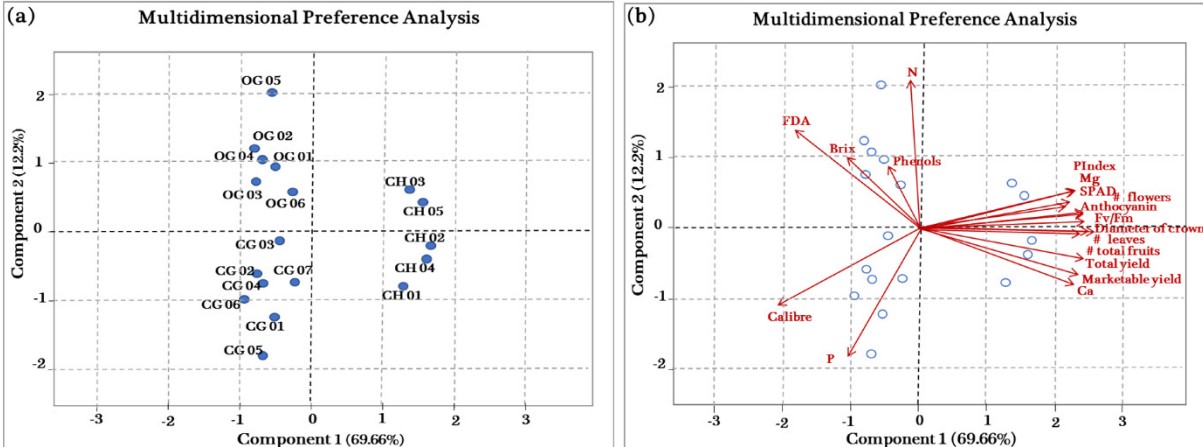

**Figure 6.** Results of the principal component analysis (PCA): projection of biostimulant treatments in the greenhouse and under the high tunnel (**a**), factor loadings for variables (**b**); soil microbial activity (FDA), performance index (PI), maximum quantum efficiency of photosystem II (Fv/Fm), chlorophyll content (SPAD), number of leaves (# leaves), number of flowering stalks (# flowers), diameter of crowns (Diameter), nitrogen (N), potassium (K), phosphorus (P), calcium (Ca), magnesium (Mg), total weight of fruits (Total yield), total number of fruits (# fruits), weight of marketable fruits (Marketable yield), fruit size (Calibre), total soluble sugars (Brix), total polyphenols (Phenol), and copper (Cu), and anthocyanin content (Antho).

Conventionally treated plants with seaweed, mycorrhiza, bacteria, and the mixture of mycorrhiza and bacteria in the greenhouse were related to a high P leaf content and fruit size. In contrast, organically grown plants treated with mycorrhiza, bacteria, the mixture of mycorrhiza and bacteria, and citric acid were associated with leaf N content, total polyphenols, sugar level, and FDA. Treatments with mycorrhiza and the mixture of mycorrhiza and bacteria in the CH were strongly related to the number of leaves, total yield, and marketable yield. In contrast, treatment with citric acid was associated with the P Index, Mg and SPAD (Figure 6b).

The main variable of PC1 was the number of fruit, followed by the total yield, marketable yield, number of leaves, diameter of the crown, Fv/Fm, number of flowering stalks, K, anthocyanins, SPAD, Mg and P Index. For PC2, the main variable was N, followed by the total polyphenols, °Brix and FDA. The Ca leaf content had the opposite relationship with the soil biological activity expressed by FDA.

## 4. Discussion

### 4.1. Impact of the Biostimulants

For all studied growing systems, biostimulant treatments had little effect on the soil microbial activity expressed by the hydrolysis of the FDA compared with their respective control (Figure 1). However, our results indicated that organic practices can significantly improve soil microbial activity. The increase in microbial diversity and activity was shown to be important for the organic matter mineralization and consequently, influences the physico-chemical properties of the soil [41]. Our results are in accordance with Araújo et al. [42] who observed that under an organic agricultural system, soil had higher microbial activity than under conventional agricultural systems. In contrast with our results, several studies reported beneficial effects of seaweed extract, *Trichoderma*, and citric acid on the microbial activity and their population in the soil. For example, Khan et al. [43] reported an enhancement in the growth of beneficial soil microbes due to the use of seaweed extracts. The reason for increasing the number and activity of microorganisms may be related to the soil structure and improvement of the moisture-holding capacity of soils treated with seaweed extracts. Our study used peat-based growing media with optimal physical and chemical properties, which may explain these different results. Furthermore, Alam et al. [44] and Spinelli et al. [45] reported the beneficial influence of seaweed extract on their bacterial population and microbial activity. Additionally, Hosseini et al. [46] reported that citric acid improved the activity of soil microorganisms. This effect could be related to the positive impact of citric acid on the mobility of phosphorous in the soil [47].

For both experiments, our investigations on strawberries showed that biostimulants did not increase the physiological parameters when expressed as Fv/Fm, PI, SPAD (Table 1), and photosynthetic parameters, such as the dark respiration rate (Rd), quantum efficiency (Φ), and maximum rate of photosynthesis (Amax) (data not shown). However, conventionally grown plants in the high tunnel had higher Fv/Fm, PI, and SPAD values when compared with the conventionally grown plants in the greenhouse. Although the benefit of using biostimulants on the photosynthetic performance was not observed in the present study, several studies reported increased chlorophyll content by using seaweed extracts. According to Spinelli et al. [45] and Fan et al. [48], seaweed extract contains betaine compounds and cytokinin-like activity, which directly affect the biosynthesis of chlorophyll. Karlidag et al. [49] reported that *Bacillus* spp. increased the chlorophyll content of strawberry leaves submitted to salt stress. Zare-Maivan et al. [50] reported similar positive effects on the chlorophyll content by using mycorrhiza (vesicular-arbuscular mycorrhiza) on maize. Nitrogen-fixing bacteria significantly increased the chlorophyll content and uptake of macro-and micronutrients in tomato and red pepper [51].

Similarly, biostimulant treatments did not significantly increase the growth parameters of greenhouse plants (Table 2). In contrast to our expectation, organically grown plants treated with seaweed extract reduced the diameter of crowns compared with their respective control. Under high tunnels, however, the treatments with citric acid, bacteria, and the mixture of mycorrhiza and bacteria outperformed the control treatment in terms of the number of flowering stalks, while bacteria increased the crown diameter (Table 2). Moreover, all biostimulants increased the dry shoot plant biomass by 55% in average. In agreement with our results, Talebi et al. [52] and Hajreza et al. [53] reported an increase in the number of flowers and the diameter of flowers for ornamental plants (*Rosa hybrida* L. and *Gazania rigens* L.) by spraying organic acids, such as citric acid, malic acid, and salicylic acid. Additionally, several studies reported the positive effects of citric acid on increasing the plant height in dill [54], stem diameter, and the number of leaves in maize [55]. According to El-Yazal et al. [55], citric acid has an antioxidant effect, which could improve cell division and protect plant cells against free radicals. In addition to these studies, Backer et al. [56] reported the effect of plant-growth-promoting rhizobacteria on the enhancement of plant growth. It was also reported that the growth parameters of vegetable crops were improved by using beneficial bacteria [23] and mycorrhiza [57,58]. In

addition, several studies showed the significant positive effects of seaweed extract [45,59] on strawberry plants cv Queen Elisa.

Some of our studied biostimulants improved the leaf N content in the greenhouse and high tunnel experiments (Table 3), which is in agreement with studies reporting positive effects of nitrogen-fixing bacteria on the nutrient uptake of different crops [60,61]. For example, Egamberdiyeva [62] reported that maize plants treated with bacteria such as *Bacillus* spp. had higher N, P and K uptake efficiency in nutrient-deficient calcisol soil. Oliveira et al. [63] reported higher N levels in the maize leaf when plants were inoculated with *A. brasilense*. The inoculation of barley plants with bacteria also increased soil and plant N concentrations [64]. Moreover, high concentrations of N, P and K were observed in maize plants sprayed with a combination of citric acid and micronutrients (Fe, Mn, and Zn) [55]. These different results compared with our study may be explained by optimal growing conditions and lower abiotic stresses under-protected cultivation, which might have mitigated the positive impact of biostimulants reported in the literature.

For conventionally grown plants in the greenhouse, citric acid increased the total number of fruits and the total and marketable yields compared with the control (Figure 2), while the total number of fruits was increased for organically grown plants. This gain of yield and fruit number was positively correlated ($p < 0.05$) with the number of leaves, the number of flowering stalks and the diameter and number of crowns. Our results are in line with the findings of El-Yazal [55] who reported that citric acid increased the yield in terms of the number and weight of grain in maize. Additionally, Abd-Allah et al. [65] reported that applying citric acid increased the plant height, yield, and protein content of the common bean, pea, and faba bean. Beneficial effects of the citric acid on the yield components were reported by Abido et al. [66] in sugar beets and Fawy and Atyia [67] in wheat, but little information has been reported about the mechanism of citric acid effect on plant productivity.

The quality attributes exhibited a significant difference between the treatments in both the greenhouse and high tunnel experiments. We hypothesized that fruit quality would be improved in the treatments with biostimulants compared with the control. In the greenhouse experiment, conventional treatment with *Trichoderma* and a mixture of mycorrhiza and nitrogen-fixing endosymbiosis bacteria in the organic part significantly increased fruit quality parameters, such as °Brix, and/or total polyphenols, and/or anthocyanins in the fruits. In the high tunnel, all biostimulant treatments increased the anthocyanin concentration of berries compared with the control, but no significant improvement was observed for °Brix and total polyphenols. Our results regarding the effect of *Trichoderma*, mycorrhiza, bacteria, and citric acid on fruit quality are in line with several findings. For example, Lingua et al. [68] reported that inoculation with arbuscular mycorrhizal fungi and plant growth-promoting *Pseudomonads* increased the anthocyanin concentration of strawberry plants cv Selva under reduced fertilization. Additionally, fruits inoculated with mycorrhiza significantly increased the quantity of glucose in tomato plants compared with other treatments [69]. Todeschini et al. [70] showed that inoculation with plant-growth-promoting bacteria increased the sugar and anthocyanin concentrations of strawberry plants cv Elyana. Pascale et al. [71] reported that the application of *Trichoderma harzianum* improved the total amount of polyphenols and antioxidant activity in grapes. Overall, individual positive effects of biostimulants on the biosynthesis of sugar in different plants were reported by several authors [72,73]. They mentioned that the higher biosynthesis of sugar could be associated with the higher chlorophyll content, chlorophyll fluorescence, net photosynthesis, and photosystem II efficiency. In our study, however, biostimulants had no positive impact on these parameters and no correlation between these parameters and °Brix values were observed ($p < 0.05$).

### 4.2. Impact of the Growing Systems

When we compared the organic and conventional growing systems, higher microbial activity was observed under organic management compared with the conventional one

(Figure 1). This increase in microbial activity may be explained by the use of organic manure containing a high content of organic nitrogen, which increased the availability of organic carbon in the growing medium. Our results agree with several studies that reported positive effects of organic fertilizers on the microbial activity of the growing medium [74,75].

The plant physiological parameters PI and Fv/Fm were different between the organic and conventional growing systems. Strawberry plants grown organically had higher PI and Fv/Fm compared with the conventionally grown plants. In contrast with our results, several studies reported the enhancement of the chlorophyll content of leaves in organic farming. For example, Macit et al. [76] reported that, the chlorophyll content of organically grown strawberry plants (cv Sweet Charlie) was higher compared with that of the conventionally grown ones. This could be explained by the fact that, under a greenhouse environment, all growing parameters are optimized compared with field experiments.

Regarding the growth parameters, no significant difference was observed between the conventional and organic greenhouse growing systems. Conventional management, however, resulted in a higher fruit yield (+16% total yield; +20% marketable yield) compared with organically grown plants (Figure 2). Macit et al. [76] also observed a higher yield of conventionally grown strawberry plants (cv Sweet Charlie) than organic ones. Additionally, Conti et al. [77] showed that strawberry plants (cv Camarosa) grown in an organic farming system produced 50% lower yield per plant than conventional strawberries. Several studies agree with the yield gap between organic and conventional agriculture [7]. The main cause of the lower yield of organically grown strawberries may be related to the limited nutrient availability, which reduces plant growth and productivity. In our study, although leaf N content was higher under organic management, lower P and Ca leaf content were observed, which were positively correlated to yield, while K was positively correlated to Pi and SPAD values ($p < 0.05$).

There was no significant difference in the fruit quality parameters between the growing systems (Figure 5), which agrees with several reviews [7,78,79]. However, some studies have reported higher fruit quality in organic farming of strawberries compared with conventional [7,8]. For example, Andrade et al. [80] reported that the °Brix of organic strawberries was 61.6% higher than that of conventional strawberries. Similarly, Oliveira et al. [63] showed that the soluble solid content of conventional tomato fruits was 56% lower than that of organic fruits. Additionally, Kobi et al. [81] reported that phenolic compounds, anthocyanin concentration and total soluble solids were higher in organically grown strawberry plants than in conventional ones. In another study, Krolow et al. [82] observed higher °Brix and anthocyanins in organic than conventional strawberries. Increasing gustatory and health components of fruits under organic farming are often related to stress conditions that increase secondary metabolites [7,78,79,83]. The difference observed between our work and published studies may be explained by the fact that the plants grown in the greenhouse had the same environmental growing conditions.

## 5. Conclusions

In the present study, we studied the impact of different biostimulants under organic and conventional crop management in a greenhouse and a high tunnel. Our results showed that biostimulants, such as bacteria, mycorrhiza, a mixture of bacteria and mycorrhiza, and citric acid are promising biostimulants in terms of plant agronomic performance, yield, and quality attributes compared with untreated plants. Many different responses of biostimulants between organic and conventional growing management, as well as between greenhouse and high tunnel growing systems were observed in this study. These differences may be related to the initial soil biological properties (e.g. organic vs. conventional) and the presence of abiotic stresses that may have occurred (e.g. temperature, light, and soil water content). Our results may have a significant impact on the berry industry by proposing a sustainable approach to improve plant growth, crop productivity, and fruit quality attributes in terms of taste and health-beneficial secondary

metabolites. However, the different results observed in the performance of biostimulants for plants grown under greenhouses and high tunnels indicate the need for further studies, which should be conducted over several growing seasons and under different abiotic conditions. Optimizing the application dose, frequency, and method (foliar, drench, or soil applications) is also essential. It should also consider the cost and profitability of biostimulants for both growing systems.

**Author Contributions:** Conceptualization, V.S. and M.D.; writing—original draft preparation, V.S.; review and editing, M.D.; statistical analysis, A.B.; resources, L.G. All authors have read and agreed to the published version of the manuscript.

**Funding:** This research was funded by the Natural Sciences and Engineering Research Council of Canada–Collaborative Research and Development (NSERC-CRD) grants #484179-2015, Les Fraises de l'Île d'Orléans Inc. and Les Tourbières Berger Ltée.

**Institutional Review Board Statement:** Not applicable.

**Informed Consent Statement:** Not applicable.

**Data Availability Statement:** The data presented in this study are available on request from the corresponding author.

**Acknowledgments:** The authors would like to thank Annie Van Sterthem, Thi Thuy An Nguyen, Louis Gauthier, Yves Gauthier, Marc Charland, Marc Bourgoin, and Laura Thériault for their collaboration in this study.

**Conflicts of Interest:** The authors declare no conflict of interest.

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
