# Peer review of "Biostimulants Promote Plant Development, Crop Productivity, and Fruit Quality of Protected Strawberries"

_agronomy, doi:10.3390/agronomy12071684_

Round 1

Reviewer 1 Report

Dear authors,

Thank you for the opportunity to meet the manuscript entitled: "Biostimulants promote plant development, crop productivity, and fruit quality of protected strawberries". The application of biostimulants in the production of field and garden crops is widespread worldwide. However, the importance of this approach is especially growing in the context of ongoing climate change, as the positive impact on the physiological and production parameters of plants under stress conditions has been presented many times.

The manuscript is written at a good level, I especially appreciate the number of monitored strawberry traits and also the number of treatment variants. In this context, however, I consider it a slight flaw that the same treatment options were not monitored in the individual systems.

In addition, it is very difficult to objectively assess the impact of treatment in green-house and high tunnel conditions, as homogeneity of conditions has not been ensured.

Introduction:

Following the topic of manuscript, I recommend adding information about biostimulants. There is a lack of information about basic groups, application options and the like. As the experiment also focused on this research, I consider it important.

Material and methods:

Developed at a decent level, the individual procedures have been clearly described. However, I recommend adding methods for determining individual elements in subchapter 2.4.5. In addition, it would be useful to specify at what stage the leaves were removed for analysis.

The various physiological parameters were reported to have been measured 3 times during the season. However, I recommend completing the phases in which they were measured. In addition, why the authors chose measurement in these phases. Whether fluorescence, chlorophyll content and others fluctuate depending on the growth phase, part of the day, but especially in the case of foliar preparations also on the date of application.

The results:

The results are well analysed, but I recommend increasing the quality of the figures. It is advisable to distinguish the individual treatment variants from each other in order to increase the clarity of the figures.

Discussion:

The partial results of the experiment are well discussed with adequate literature sources.

Conclusions:

701 As mentioned in chapter MM, the conditions (light, heat, etc.) were the same for all variants, so I do not consider this conclusion to be correct.

Reviewer 2 Report

In this paper, the authors conducted a very interesting large-scale experiment, testing different biostimulants in different cultivation systems of strawberries. The measurements were also carefully selected to provide new knowledge about the impact of biostimulants. The paper is correctly designed and carried out. Results are also consistent and very interesting. However, the current version must be revised at several points before it can be accepted. A list of the points to be addressed when revising the paper is provided below:

 L43-45: Please re-write the sentence in a more understandable way

L57: I propose this sentence to be written as follows: “Biostimulants are applied exogenously and are considered as substances to plants…” Please let me know if this is what you meant here.

L60: Biostimulants can indeed increase resilience to biotic stresses. However, I would suggest you to make here a statement that according to EU laws, biostimulants need to be clearly separated with PPPs (Plant Protection Products). For more information, you can have a look at du Jardin (2015), an article already referred at your paper. If you do not wish to add this information, you can ignore this specific comment.

L155: Please write 4 oC and not four oC

L214: Once or twice a week, not weekly

L219: Please re-write the sentence

L224: Why did you have different number of measurements between the greenhouse and the high tunnel ?

L317: Replace parameter with parameters

L326: I did not understand exactly the light response curves mentioned here. Can you please inform me where is this addressed at the Material and Methods?

Figure 2b: Replace weeks with week

Figure 2. Fruit size differs in contrast with what is said in the text at L459-460

Figure 2. Can you please provide more information about what is measured here? When you say approximately 40 grams of fruits per week, do you mean per plant? Per treatment? It would be interesting to add maybe inside the text, the yield per m2 (if not for each treatment, at least the avg yield of the cultivated systems (organic, conventional, high tunnel)

L651-653: Sentence is not aligned with the results of the experiment. You did not observe significant differences for chlorophyll content, chlorophyll fluorescence, net photosynthesis, however you had significant differences at oBrix.
